# Are Plants Capable of Pheomelanin Synthesis? Gas Chromatography/Tandem Mass Spectrometry Characterization of Thermally Degraded Melanin Isolated from *Echinacea purpurea*

Slawomir Kurkiewicz [1], Łukasz Marek [1], Małgorzata Kurkiewicz [2], Adam Kurkiewicz [3] and Anna Dzierżęga-Lęcznar [1,*]

1 Department of Instrumental Analysis, Faculty of Pharmaceutical Sciences in Sosnowiec, Medical University of Silesia, 40-055 Katowice, Poland
2 Platomics, 1020 Vienna, Austria
3 BioMage Ltd., Edinburgh EH4 2HS, UK
* Correspondence: adler@sum.edu.pl

**Abstract:** *Echinacea* is a widely used plant medicine, valued especially for its well-documented ability to stimulate the immune system. It has been suggested that melanin could be one of the bioactive factors responsible for the immunostimulatory properties of the plant. The biological functions of melanin pigments are closely related to their chemical composition and structural features. The aim of this study was to characterize the melanin from *Echinacea purpurea* based on the analysis of thermal degradation products of the well-purified pigment extracted from the dried herb. The melanin was pyrolyzed, and the resulting products were separated by gas chromatography and identified using a triple quadrupole mass spectrometer operating in full scan and multiple reaction monitoring modes. Three groups of marker products were detected in the melanin pyrolysate: polyphenol derivatives, nitrogen-containing heterocycles devoid of sulfur, and benzothiazines/benzothiazoles. This suggests that *E. purpurea* produces three structurally different melanin pigments: allomelanin, eumelanin, and pheomelanin, which in turn may affect the biological activity of the herb. Our results provide the first-ever evidence that plants are capable of synthesizing pheomelanin, which until now, has only been described for representatives of the animal and fungal kingdoms.

**Keywords:** immunostimulatory activity; pheomelanin; *Echinacea purpurea*; Py-GC/MS/MS; Py-GC/MS

## 1. Introduction

*Echinacea purpurea* is one of the best-known and relatively safe medicinal plants. It has been traditionally used to treat and prevent cold, flu, and other upper respiratory tract infections [1,2]. In self-medication, the plant is usually taken in the form of aqueous or alcoholic extracts (infusions or tinctures) of the dried aerial parts or rhizome and roots. It is also commercially available as standardized preparations in solution or capsule form, often in combination with ginseng, goldenseal, or garlic [3,4]. *Echinacea* preparations belong to the group of best-selling botanical drugs in the USA and Europe [5]. Dietary supplements to treat the common cold based on *Echinacea* species are among the most frequently used natural products in the USA [4,6]. Numerous studies have shown that the therapeutic effect of *Echinacea* is related to the anti-inflammatory and antioxidant properties of the plant and its direct antibacterial and antiviral activity, but most importantly, to its strong stimulation of the immune system. Through activation of the neutrophils, macrophages, polymorphonuclear leukocytes, and natural killer cells, both innate and adaptive immunity is enhanced by administration of the plant preparations [1,4].

Many plants that are traditionally used to enhance immune functions in humans have been reported to contain melanin or "melanin-like" material. Melanins are a heterogeneous group of polymeric pigments distributed in most living organisms. They are formed via the enzyme-catalyzed oxidation of various precursor compounds, which explains the high structural diversity of the resulting polymers. Based on the structure of monomer units, melanin pigments can be divided into three main types: Eumelanins are composed of indole-type units derived from tyrosine or L-DOPA (L-3,4-dihydroxyphenylalanine). Pheomelanins consist of sulfur-containing units of 1,4-benziothiazine and 1,3-benzothiazole type that arise from the oxidation of the cysteinyl conjugates of L-DOPA. The third type of melanin pigments, allomelanins, are the most heterogeneous structurally, and comprise biopolymers derived from phenol nitrogen-free precursors, such as catechol or catecholic acids (caffeic, chlorogenic, protocatechuic, or gallic acids), dihydroxynaphthalene or other types of dihydroxybenzenes [7,8]. Melanins have a wide variety of functions in living organisms. Some of these have yet to be studied, but most are related to protection from external factors, providing melanized organisms with an environmental advantage [7,9,10].

There is some evidence that melanin may be one of the factors responsible for the immunomodulatory properties of various plant products consumed by humans [9,11]. Plant melanins were found to be recognized by the toll like receptors (TLR) family, thereby inducing an innate immune response. For instance, melanin isolated from *Echinacea* was able to activate cultured monocytes by binding to TLR-2, which led to the increase in the cytokine IL-1β secretion by the cells in a dose-dependent manner. Furthermore, in vivo experiments showed that the oral intake of melanin derived from *Echinacea* species by mice enhanced the production of IL-6 in Peyer's patch cells, and IFN-γ in spleen cells [12].

The biological functions of melanin pigments appear to be closely related to their chemical composition and structural features. For example, in addition to their well-known protective role against UV radiation, animal eumelanins are potent antioxidants and effective free radical scavengers, while pheomelanins are considered photosensitizers, with pro-oxidant activity [13–15]. Our recent in vitro studies on human melanocytes suggest that the ability of these immunocompetent skin cells to secrete cytokines and chemokines in response to inflammatory stimuli via a TLR-dependent signaling pathway may be significantly modified by the level and structural features of cellular melanin [16].

The aim of this study was to characterize the structural components of melanin from *Echinacea purpurea* by gas chromatography/mass spectrometry (GC/MS) and gas chromatography tandem mass spectrometry (GC/MS/MS) analysis of the products formed during the thermal degradation (pyrolysis, Py) of the pigment. Pyrolysis, in combination with gas chromatography and mass (or tandem mass) spectrometry, allows for rapid and efficient differentiation of melanin types and evaluation of non-melanin pigment components, and for these reasons, it is considered a valuable tool for melanin analysis. The technique has been successfully applied to characterize melanin pigments isolated from diverse biological sources, such as bacteria, soil fungi, insects, bird feathers, and human hair, skin, or brain tissue [9,17].

## 2. Materials and Methods

### 2.1. Reagents and Compounds

L-tyrosine, mushroom tyrosinase (EC 1.14.18.1; 6680 U/mg of solid), ammonium hydroxide, hydrochloric acid (35–38%), isopropanol, and hexane were purchased from Sigma–Aldrich Inc (St. Louis, MO, USA), and were of the highest purity available.

### 2.2. Preparation of Melanin Standard

Reference eumelanin was prepared by the tyrosinase-catalyzed oxidation of L-tyrosine (Tyr-melanin, eumelanin standard). Melanin precursor was dissolved in a phosphate buffer (50 mM, pH 6.8) to obtain the final concentration of 10 mM, then 100 U tyrosinase/mL was added, and the reaction mixture was incubated for 48 h at 37 °C, with vigorous stirring and

protection from light. The pigment formed was collected by centrifugation, washed several times with deionized water, and dried to a constant weight at 37 °C.

### 2.3. Isolation of Melanin from Echinacea purpurea

The method for isolation and purification of melanin from plant material was developed on the basis of the method described by Sava et al. [18]. Dried echinacea herb, with an organic certificate, was used in the research (*Echinacea purpurea* Dary Natury, Gdańsk, Poland). A total of 75 mL of 2% aqueous $NH_4OH$ solution was added to 10 g of ground echinacea herb, and the mixture was incubated at room temperature in a shaking water bath for 24 h, then acidified with 2 M hydrochloric acid to pH 2.5 and further incubated for 2 h. The resulting crude precipitate was centrifuged (10 min, 4500 *g*). In order to remove protein and polysaccharide contaminants, 40 mL of 6 M hydrochloric acid was added to the precipitate and the mixture was heated for 2 h at 100 °C. The resulting suspension was then centrifuged (10 min, 4500 *g*), and the pellet was washed six times with deionized water. To remove possible lipid contamination, an additional washing procedure was carried out with isopropanol (twice), isopropanol:hexane (1:1, *v/v*), and hexane. The purified melanin isolate was dried to a constant weight at 37 °C.

### 2.4. Conditions of Py-GC/MS and Py-GC/MS/MS Analysis

Melanin pigments were thermally degraded at 500 °C using a microfurnace-type pyrolyser (Pyrojector II, SGE Analytical Science) with a P3 Pelletiser Solids Injector (SGE Analytical Science). The pyrolysis products were transferred with a stream of helium (155 kPa) via 0.12 mm transfer tube directly to the split/splitless injector of the Agilent Technologies 7890A gas chromatograph (splitless time: 0.3 min). The GC separations were performed on an RTX-5MS capillary column (5% diphenyl, 95% dimethylpolysiloxane, 60 m × 0.32 mm i.d. ×0.5 μm film thickness) with helium (96.5 kPa) as the carrier gas. The GC oven temperature was programmed from 35 °C (held for 5 min) to 100 °C at a rate of 5 °C/min, then to 260 °C at a rate of 10 °C/min. The final temperature was constant for 16 min. The GC column outlet was connected directly to the ion source of the Agilent Technologies 7000 GC/MS Triple Quad mass spectrometer. The temperatures of GC/MS interface, the ion source, and the quadrupoles were 240 °C, 230 °C, and 150 °C, respectively. An electron ionization (70 eV) was applied. The tandem mass spectrometer was operated in a full scan (MS1 scan, m/z 45–450) or a multiple reaction monitoring (MRM) mode. Nitrogen (constant flow rate: 1.5 mL/min) and helium (2.25 mL/min) were used as the collision and the quench gas, respectively. The list of the thermal degradation products that were monitored in the melanin pyrolysates by tandem MS, along with the optimal MRM settings, are shown in the Supporting Information section (Table S1). Before a series of Py-GC/MS and Py-GC/MS/MS analyses, thermochemolysis was performed to confirm the purity of the melanin isolated from *E. purpurea*. For this purpose, a drop of 10% methanolic solution of tetramethylammonium hydroxide (TMAH) was placed on the top of a sample aliquot compressed in the pelletizer, and the sample pyrolysis was performed under the conditions described above. The mixture of compounds formed as a result of the thermally assisted hydrolysis and methylation were checked for the presence of derivatized breakdown products of protein, carbohydrate, and lipid origin. MassHunter GC/MS Acquisition B.07.01 and MassHunter Workstation Qualitative Analysis B.07.00 (Agilent Technologies, Santa Clara, CA, USA) software were used for data collection and mass spectra processing. Pyrolysis products were identified by the comparison of their mass spectra with the library standards (the NIST/EPA/NIH Mass Spectral Library 2011 and the Wiley Registry of Mass Spectral Data 14th Edition) and by comparison of the Kovats retention indices (RI) calculated on RTX-5MS column with the tabulated values (NIST).

### 3. Results and Discussion

Plant melanins are the most heterogeneous and the least understood group of melanin pigments, traditionally classified as allomelanins. They can be formed from a wide variety

of precursors, whose common structural feature is the absence of nitrogen [7]. The key enzymes for allomelanin biosynthesis are polyphenol oxidases (PPOs), which oxidize precursors to readily polymerizable quinone derivatives. [19,20]. It was long thought that allomelanin synthesis was related only to the enzymatic browning reaction occurring in damaged plant tissues. Due to the destruction of the cell structure, the chloroplast PPO is released, which interacts with their vacuolar substrates to form o-quinones, which in turn polymerize into melanin [21]. However, recent studies show that this is not the only mechanism for the formation of plant melanin, and that the pigment can also be produced in undamaged plant cells and deposited in chlorophyll-containing plastids [20].

Since the object of our interest in this study was the structure, but not the biological activity, of melanin, we decided to use a relatively harsh procedure based on aggressive reagents for its isolation from the plant material. The protocol used yielded a well-purified pigment, free of proteins, polysaccharides, and lipids, as verified by the TMAH thermochemolysis. The pigment was then subjected to thermal degradation in an inert gas atmosphere (pyrolysis). Pyrolysis of macromolecules, including melanin, generates a wide range of volatile products. The GC chromatogram of these products (pyrogram) is a "fingerprint feature" of the pigment molecule, and—of particular importance—allows for the rapid detection of marker compounds for a given structural type of melanin. As shown in Figure 1A and Table 1, the pyrolysate of melanin from *E. purpurea* was dominated by nitrogen-devoid polyphenol derivatives. Based on the mass spectra and the Kovats retention indices, the most abundant products were identified as catechol (retention time: 24.52 min), 2-methoxy-4-vinylphenol (retention time: 26.74 min), and 4-ethylcatechol (retention time: 27.82 min). In the pyrogram of synthetic melanin obtained from tyrosine, shown for comparison in Figure 1B, benzyl nitrile and large amounts of pyrrole, pyridine, and indole derivatives were identified, in addition to the dominant phenol. The nitrogen-containing compounds are the most characteristic products of thermally degraded eumelanins. All those compounds were also detected in the pyrolysate of melanin from *E. purpurea*, but their levels were substantially lower. The profile of the pyrolysis products, which—in contrast to the profile of pure eumelanin from tyrosine—is dominated by polyphenolic compounds, clearly indicates that the pigment isolated from *E. purpurea* mainly consists of allomelanin, with a small addition of eumelanin (Figure 2).

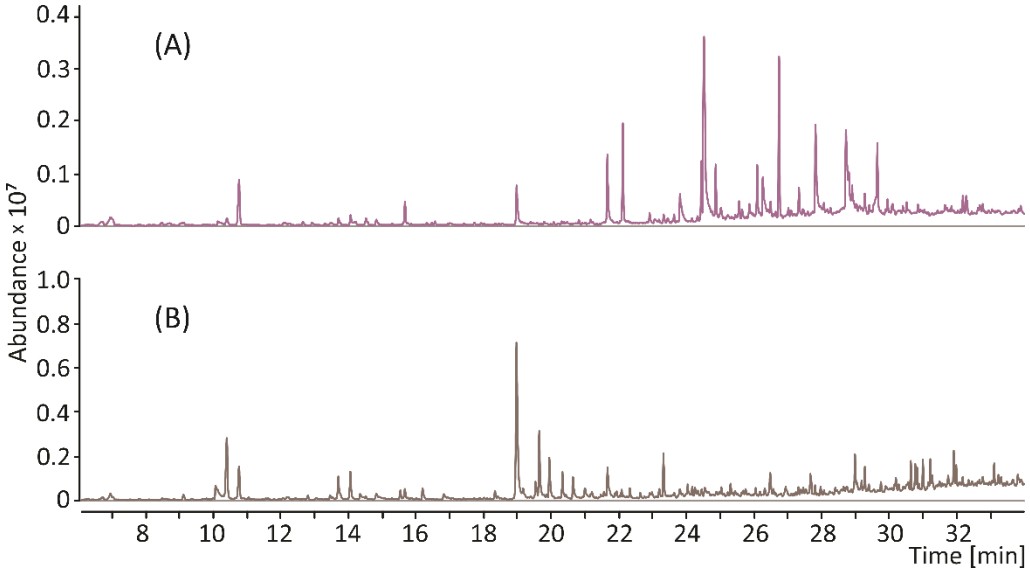

**Figure 1.** Total ion current chromatograms of the thermal degradation products of natural melanin isolated from *Echinacea purpurea* (**A**), and eumelanin from tyrosine (**B**).

**Table 1.** Characteristics of pyrolysis products of melanin from *Echinacea purpurea* and their comparison with pyrolysis products of the reference eumelanin.

| RT [min] | Compound | Group | RI RTX-5MS | RI NIST | CAS | Ech-mel % | Tyr-mel % |
|---|---|---|---|---|---|---|---|
| 6.97 | Benzene | B | 661 | 661 | 71-43-2 | 1.65 | 1.91 |
| 10.76 | Toluene | B | 767 | 767 | 108-88-3 | 5.10 | 6.68 |
| 14.52 | Ethylbenzene | B | 865 | 866 | 100-41-4 | 0.65 | 0.51 |
| 14.82 | m/p-Xylenes | B | 873 | 873 | 108-38-3/ 106-42-3 | 0.61 | 1.48 |
| 15.67 | Styrene | B | 895 | 895 | 100-42-5 | 2.00 | 1.90 |
| 15.75 | o-Xylene | B | 897 | 897 | 95-47-6 | 0.10 | 0.19 |
| 23.63 | Benzene, 1,2,3,4-tetramethyl- | B | 1166 | 1169 | 488-23-3 | 0.50 | - |
| | | **B** | | | | 10.61 | 12.67 |
| 23.33 | Benzyl nitrile | BN | 1151 | 1150 | 140-29-4 | 0.67 | 6.17 |
| 25.44 | Benzenepropanenitrile | BN | 1257 | 1250 | 645-59-0 | 0.16 | 0.46 |
| | | **BN** | | | | 0.84 | 6.64 |
| 10.41 | Pyrrole | Pyrr | 758 | 758 | 109-97-7 | 0.80 | 13.27 |
| 13.7 | 1H-Pyrrole, 2-methyl- | Pyrr | 844 | 853 | 636-41-9 | 0.83 | 5.25 |
| 14.06 | 1H-Pyrrole, 3-methyl- | Pyrr | 853 | 856 | 616-43-3 | 1.05 | 5.29 |
| 17.42 | 1H-Pyrrole, 2,5-dimethyl- | Pyrr | 942 | 937 | 625-84-3 | 0.21 | 0.22 |
| 17.73 | 1H-Pyrrole, 3-ethyl- | Pyrr | 951 | 950 | 1551-16-2 | 0.22 | 0.12 |
| 20.07 | 1H-Pyrrole, 2,3,5-trimethyl- | Pyrr | 1019 | - | 2199-41-9 | 0.14 | 0.04 |
| 20.82 | 1H-Pyrrole, 2-ethyl-4-methyl- | Pyrr | 1046 | - | 69687-77-0 | 0.28 | 0.03 |
| 22.27 | 1H-Pyrrole, 3-ethyl-2,4-dimethyl- | Pyrr | 1099 | - | 517-22-6 | 0.07 | - |
| 24.33 | 1H-Pyrrole, 3-ethyl-2,4,5-trimethyl- | Pyrr | 1200 | - | 520-69-4 | 0.31 | - |
| | | **Pyrr** | | | | 3.90 | 24.23 |
| 10.05 | Pyridine | Pr | 748 | 748 | 110-86-1 | 0.86 | 6.31 |
| 12.91 | Pyridine, 2-methyl- | Pr | 823 | 821 | 109-06-8 | 0.24 | 0.23 |
| | | **Pr** | | | | 1.09 | 6.53 |
| 18.98 | Fenol | Phe | 984 | 984 | 108-95-2 | 3.52 | 39.03 |
| 21.16 | o-Cresol | Phe | 1059 | 1058 | 95-48-7 | 0.25 | 0.15 |
| 21.66 | m/p-Cresol | Phe | 1077 | 1077 | 108-39-4/ 106-44-5 | 5.43 | 5.19 |
| 23.81 | Phenol, 4-ethyl- | Phe | 1175 | 1173 | 123-07-9 | 3.13 | 0.20 |
| 24.86 | 4-Winylofenol | Phe | 1227 | 1229 | 2628-17-3 | 3.16 | 0.57 |
| 29.65 | 2-Tert-Butyl-4-isopropyl-5-methylphenol | Phe | 1518 | - | - | 6.33 | - |
| | | **Phe** | | | | 21.82 | 45.14 |
| 22.12 | Phenol, 2-methoxy- | PPhe | 1094 | 1094 | 90-05-1 | 6.08 | 0.12 |
| 24.44 | Phenol, 2-methoxy-4-methyl- | PPhe | 1206 | 1207 | 93-51-6 | 3.07 | - |
| 24.52 | Catechol | PPhe | 1210 | 1210 | 120-80-9 | 22.03 | 1.77 |
| 26.1 | Phenol, 4-ethyl-2-methoxy- | PPhe | 1291 | 1260 | 2785-89-9 | 3.09 | - |
| 26.26 | 1,2-Benzenediol, 4-methyl- | PPhe | 1299 | 1295 | 452-86-8 | 4.32 | - |
| 26.74 | 2-Methoxy-4-vinylphenol | PPhe | 1330 | 1330 | 7786-61-0 | 10.49 | - |
| 27.33 | Phenol, 2,6-dimethoxy- | PPhe | 1367 | 1367 | 91-10-1 | 1.61 | - |
| 27.44 | Phenol, 2-methoxy-4-(2-propenyl)- | PPhe | 1374 | 1374 | 97-53-0 | 0.14 | - |
| 27.82 | 4-Ethylcatechol | PPhe | 1397 | 1392 | 1124-39-6 | 8.10 | - |
| 32.28 | Phenol, 2,6-dimethoxy-4-(1E)-1-propen-1-yl- | PPhe | 1720 | 1704 | 20675-95-0 | 0.84 | - |
| | | **PPhe** | | | | 59.76 | 1.89 |
| 26.49 | 1H-Indole | Ind | 1314 | 1316 | 120-72-9 | 1.10 | 2.08 |
| 28.07 | Methylindole | Ind | 1414 | - | - | 0.33 | 0.57 |
| | | **Ind** | | | | 1.43 | 2.65 |
| 8.48 | Furan, 2,5-dimethyl- | Fur | 707 | 708 | 625-86-5 | 0.39 | 0.26 |
| 16.47 | 1-(2-Furanylo)-etanon | Fur | 917 | 917 | 1192-62-7 | 0.15 | - |
| | | **Fur** | | | | 0.54 | 0.26 |

RT—retention time; RI RTX-5MS—Kovats retention index calculated on the RTX-5MS column; RI NIST—Kovats retention index from the NIST and Wiley library; CAS—Chemical Abstracts Service Registry Number; Ech-mel—melanin isolated from *Echinacea purpurea*; Tyr-mel—reference eumelanin from L-tyrosine. Groups: benzene and benzene derivatives (B); benzyl nitrile (BN); pyrrole (Pyrr); pyridine (Pr); phenol (Phe); polyphenols (PPhe); indole (Ind); furans (Fur).

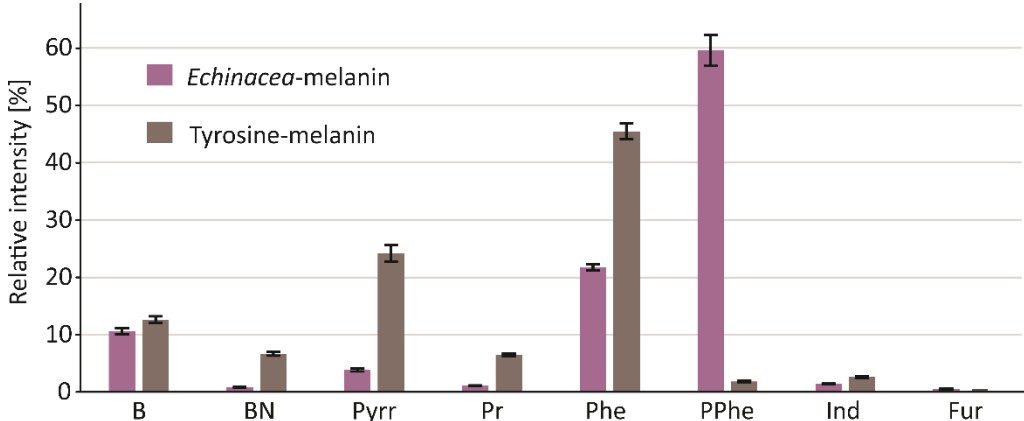

**Figure 2.** Comparison of pyrolytic profiles of reference eumelanin and melanin isolated from the echinacea herb. Groups: benzene and benzene derivatives (B); benzyl nitrile (BN); pyrrole (Pyrr); pyridine (Pr); phenol (Phe); polyphenols (PPhe); indole (Ind); furans (Fur).

Similar results, showing the presence of eumelanin structural units, were obtained by Pugh et al. for Py-GC/MS-analyzed melanin isolated from another member of the *Echinacea* family, *E. angustifolia* [12]. Interestingly, elemental analysis of the pigment showed the presence of sulfur, which could suggest either insufficient purification from residual proteins (the authors used a mild isolation procedure based on the pigment extraction with 90% aqueous phenol solution) or the presence of pheomelanin-type units in the melanin structure. Pyrolysis products of pheomelanin should contain sulfur [9], but the authors do not mention such compounds in the pyrolysate of *E. angustifolia* melanin. Detecting trace amounts of the pheomelanin markers among the pyrolysis products by GC/MS is challenging due to the limited sensitivity of the method and the lack of reference spectra in the mass spectral libraries. Our team has developed a method that allows for overcoming this problem and detecting pheomelanin in any natural melanin pigment at extremely low levels [22]. The method is based on the analysis of the pyrolytic markers of pheomelanin units by GC/MS/MS, using a triple quadrupole tandem mass spectrometer operating in the multiple reaction monitoring (MRM) mode. Such a mode provides extremely specific and sensitive detection and identification of the target molecule by the simultaneous measurement of its characteristic pair(s) of precursors and product ions (MRM transitions). The pyrolytic markers of pheomelanin-type structural units are mainly the derivatives of 1,4-benzothiazine and 1,3-benzothiazole and, contrary to the products characteristic of eumelanin-type units, are not formed during the pyrolysis of proteins or other natural substances. All those markers were detected in the pyrolysate of melanin isolated from *E. purpurea* (Figure 3, Table 2), which provides hard evidence for the presence of pheomelanin in this material.

Pheomelanin was long thought to be specific to animals, particularly mammals and birds [7]. Some researchers postulated the presence of pheomelanin in fungi (*Lachnum singerianum*), but the presence of its chemical markers, especially those containing sulfur, was not confirmed by them [23]. Recently, using EPR spectroscopy, Pukalski et al. [24] identified significant amounts of pheomelanin-like pigment in mycelium of *Plenodomus biglobosus*. We are the first to show that pheomelanin can also be synthesized by higher plants. There are many reports, based on both degradative and non-destructive analytical techniques, indicating that the same animal or fungal organism is capable of synthesizing melanin from different precursors through secondary metabolism [25,26]. In the case of animals, including humans, the term "mixed melanogenesis" has even been proposed to emphasize this phenomenon [27].

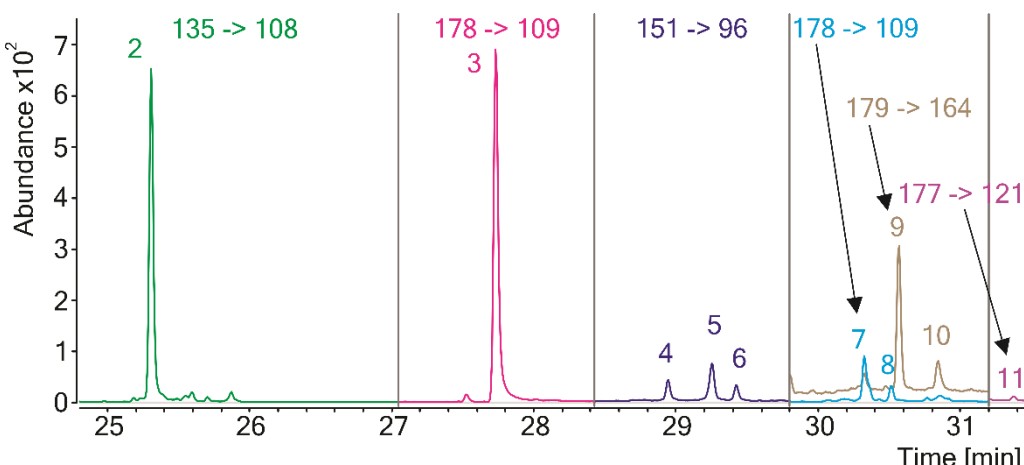

**Figure 3.** Extracted MRM chromatogram of the pyrolytic pheomelanin markers of melanin from *Echinacea purpurea*. Each peak corresponds to the most intense MRM transition. For peak designation, see Table 2.

**Table 2.** Pheomelanin markers detected among pyrolysis products of melanin isolated from *Echinacea purpurea*.

| No. | RT [min] | Compound | RI RTX-5MS | MS Transition * |
|---|---|---|---|---|
| 1 | 9.6 | Thiazole | 736.7 | 85 -> 58 |
| 2 | 25.3 | Benzothiazole | 1250.0 | 135 -> 108 |
| 3 | 27.7 | 4-Hydroxybenzothiazole | 1391.8 | 151 -> 96 |
| 4 | 28.9 | 2,3-Dihydro-5H-1,4-benzothiazin-5-one (isomer 1) | 1470.8 | 165 -> 136 |
| 5 | 29.3 | 2,3-Dihydro-5H-1,4-benzothiazin-5-one (isomer 2) | 1490.3 | 165 -> 136 |
| 6 | 29.4 | 2,3-Dihydro-5H-1,4-benzothiazin-5-one (isomer 3) | 1501.5 | 165 -> 136 |
| 7 | 30.3 | 7-Methy-2,3-dihydro-5H-1,4-benzothiazi-5-one (isomer 1) | 1566.7 | 178 -> 109 |
| 8 | 30.5 | 7-Methy-2,3-dihydro-5H-1,4-benzothiazi-5-one (isomer 2) | 1581.2 | 178 -> 109 |
| 9 | 30.6 | 4-Hydroxy-6-ethylbenzothiazole (isomer 1) | 1584.8 | 179 -> 164 |
| 10 | 30.9 | 4-Hydroxy-6-ethylbenzothiazole (isomer 2) | 1605.5 | 179 -> 164 |
| 11 | 31.4 | 7-Methyl-5H-1,4-benzothiazin-5-one | 1647.3 | 177 -> 121 |

RT—retention time; RI RTX-5MS—Kovats retention index calculated on the RTX-5MS column. * The most intense transition (m/z -> m/z).

Further research is needed to clarify the benefits of pheomelanin synthesis for *E. purpurea* (and for plants in general). It seems likely, however, that the presence of pheomelanin could be one of the factors responsible for the particularly potent immunostimulatory activity of the plant. Our previous studies on cultured human melanocytes of different pigmentation phenotypes showed that the cells with higher pheomelanin content exhibited higher TLR4 mRNA expression after stimulation with bacterial lipopolysaccharides and secreted significantly more CXCL8 (IL-8) [16,28]. This chemokine exhibits potent chemotactic activity for neutrophils, which play a key role in the innate immune response.

## 4. Conclusions

Three groups of marker compounds were detected among the pyrolysis products of melanin isolated from *Echinacea purpurea* using GC/MS and GC/MS/MS: the nitrogen-free polyphenols, the nitrogen-containing heterocycles devoid of sulfur, and the derivatives of 1,4-benzothiazine and 1,3-benzothiazole. Thus, it can be concluded from these analyses that *E. purpurea* produces three structurally different melanin pigments: allomelanin, eumelanin, and pheomelanin. To the best of our knowledge, this study provides the first-ever evidence that plants are capable of synthesizing pheomelanin, which, until now, has only been

described for representatives of the animal and fungal kingdoms. Although it is difficult to speculate on the importance of pheomelanin for the plant organism, we believe its presence may be relevant to the pharmacological activity of *E. purpurea*.

**Supplementary Materials:** The following supporting information can be downloaded at: https://www.mdpi.com/article/10.3390/pr10112465/s1, Table S1: Multiple reaction monitoring (MRM) settings for the analysis of melanin pyrolysate by GC/MS/MS.

**Author Contributions:** Conceptualization, S.K., A.D.-L. and Ł. M.; methodology, S.K. and A.D.-L.; investigation, S.K., A.D.-L., Ł.M., M.K. and A.K.; writing–original draft, S.K., A.D.-L. and Ł.M.; writing–review and editing, A.D.-L., M.K. and A.K.; visualization, S.K.; supervision, A.D.-L.; project administration, A.D.-L. All authors have read and agreed to the published version of the manuscript.

**Funding:** This research received no external funding and the APC was funded by S.K.

**Conflicts of Interest:** The authors declare no conflict of interest.

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
