# Peer review of "Are Plants Capable of Pheomelanin Synthesis? Gas Chromatography/Tandem Mass Spectrometry Characterization of Thermally Degraded Melanin Isolated from Echinacea purpurea"

_processes, doi:10.3390/pr10112465_

Round 1

Reviewer 1 Report

The present study gives new insights concerning structure of plant melanins. To date, melanins in plants are insufficiently explored area, which has a great potential for pure and applied studies. The manuscript leaves a pleasant impression and, in my opinion, can be accepted in present form.

Author Response

We agree with the Reviewer's opinion that plant melanins are still insufficiently researched and research in this area has a great scientific potential. We thank the reviewer for their time and effort required to review our work. 

Reviewer 2 Report

1. What is the molecular weight of the purified plant melanin?

2. How the purity of the extracted melanin was confirmed? Is there any possibility to perform chromatography (TLC or HPLC) to evaluate the purity of the extracted melanin so that the probability of any impurity would be ruled out?

3. There are previous reports on the presence of pheomelanin in plants that needs to be cited in the manuscript.

4. NMR analysis of the extracted melanin is required to correlate the obtained result from the pyrolysis to the melanin pigments.

5. Is there possibility that the plant contain all three types of melanins (pheomelanin, eumelanin, and allomelanin) and not that single melanin with three components from pheomelanin, eumelanin, and allomelanin.

6. It is highly unlikely that a single melanin polymer contains components of pheomelanin, eumelanin, and allomelanin. Need additional data to prove this claim. Is there any such previous report in the literature?

7. Authors are suggested to perform UV-VIs, EPR, SEM-EDX experiments to get additional ideas about the structure of the extracted melanin. Furthermore, this data will be helpful in identifying the correct class of the extracted melanin.

8. Dialysis of the extracted melanin may be performed to remove impurities. 

Round 2

Reviewer 2 Report

It is good to see the modification done to the original manuscript which is now much better. However, my main concern regarding the identification of melanin type on the basis of the pyrolysis-GC-MS/MS method remains unanswered. Since it is a degradation method, we can not be 100% sure about the conclusion. I agree with the authors that this is the commonly employed method for the identification of melanin pigments. However, it has its limitations, and the use of this method to conclude that extracted melanin contains components of allomelanin, eumelanin, and pheomelanin is an overstatement. Degradation methods were routinely used for the determination of structures of natural products a few decades back before the advent of advanced NMR and MS techniques, but now no one employs the degradation methods for natural products. But in the case of melanin, there is ambiguity in structure due to the absence of a proper method. Similarly, due to the absence of the complete structures of melanin, I would not go ahead with the author's conclusion that the extracted melanin contains all three types of melanins. Therefore, two things are suggested; first confirmation of the purity of the melanin, which is not possible at the moment can be understandable due to the insolubility of melanin in almost all solvents. Secondly, the authors are suggested to perform an elemental analysis of the melanin by SEM-EDX, and mention the data in the manuscript to support the claim about the presence of three types of melanin. The elemental analysis will at least give us an idea about the percentage of C, O, N, S in the extracted melanin. Authors are further suggested to cite such published papers where extracted melanins from plants, microbes, or animals are found to be a mixture of three types of melanins where the pyrolysis-GC-MS/MS (destructive) method was not used. To summarize, I would say that the conclusion of the manuscript that the extracted melanin contains all three types of melanin pigments is correct as per the employed pyrolysis-GC-MS/MS method, but it may not be 100% correct if other methods are used.   
